# Association Between 24-Hour Movement Behaviors and Noncommunicable Chronic Diseases Among Adult and Older Adult Users of the Brazilian Community Health Promotion Program

**DOI:** 10.3390/healthcare13162016

**Published:** 2025-08-15

**Authors:** Yuri Silva de Souza, Carlos Alencar Souza Alves Junior, Diego Augusto Santos Silva

**Affiliations:** 1Graduate Program in Physical Education, Department of Physical Education, Sports Center, Federal University of Santa Catarina, Florianópolis 88040-900, SC, Brazil; yurisilvaedf@gmail.com; 2Center for Integration of Teaching, Research, and Extension in the Area of Languages and Their Technologies, Department of Languages, Federal Institute of Education, Science and Technology of Rio Grande do Sul, Canoas Campus, Canoas 92412-240, RS, Brazil; alvesjunior.cas@gmail.com

**Keywords:** physical activity, screen time, sleep

## Abstract

**Background/Objectives:** Noncommunicable diseases (NCDs) are the leading causes of global mortality among adults. The aim of this study was to examine the association between adherence to 24 h movement behavior guidelines and the diagnosis of NCDs. **Methods:** This cross-sectional study was conducted with users of the Health Academy Program in Brazil. The sample consisted of 1212 individuals (92.9% female), aged 18 years or older. Dependent variables included self-reported hypertension, diabetes, hypercholesterolemia, and cardiovascular disease based on previous medical diagnosis. Independent variables (physical activity, screen time, and sleep) were self-reported. Binary and multinomial logistic regressions were performed and adjusted for sex, age, educational level, body mass index, and marital status. **Results:** Participants who did not meet any of the 24 h movement behavior recommendations had higher odds of hypertension (OR: 1.35; 95% CI: 1.15–1.77), diabetes (OR: 1.07; 95% CI: 1.03–2.01), and having two (OR: 1.29; 95% CI: 1.09–2.91) or three or more NCDs (OR: 1.40; 95% CI: 1.11–2.13). Not meeting the physical activity recommendation was associated with higher odds of hypercholesterolemia (OR: 1.37; 95% CI: 1.06–1.76). In contrast, meeting the physical activity guideline alone (OR: 0.32; 95% CI: 0.11–0.85) or in combination with adequate sleep (OR: 0.32; 95% CI: 0.11–0.90) was associated with lower odds of cardiovascular disease. All of these results remained significant after adjustments for multiple comparisons. **Conclusions:** Not meeting any of the 24 h movement behavior guidelines, especially those related to physical activity, was associated with a higher occurrence of NCDs.

## 1. Introduction

Noncommunicable diseases (NCDs) constitute a significant public health challenge. In 2021 alone, they accounted for approximately 19 million deaths across all age groups [1]. Globally, an estimated 37.2% of the population is affected by NCDs; however, in South America, the prevalence is even higher, reaching 45.7% [2]. Low- and middle-income countries are experiencing a significant increase in the prevalence of type 2 diabetes [3] and multimorbidity [4]. The burden of NCDs is decreasing in high-income countries; however, the same trend is not observed in low- and middle-income countries [5]. In Brazil, NCDs were associated with 734,000 deaths in 2019, representing 55% of all fatalities in the country [3]. This burden contributes substantially to increased healthcare expenditures [4].

Physical activity (PA) [5], screen time [6], and sleep duration [7] have been shown to influence the development of NCDs in adults and older adults. Adequate levels of PA are associated with reduced mortality risk, whereas excessive screen time increases the likelihood of developing NCDs and of premature death [5,6]. Furthermore, shortened sleep duration has also been linked to a higher risk of mortality in adults [7]. These factors are associated not only with an increased risk of developing NCDs, but also with higher all-cause mortality rates [5,6,7].

Historically, lifestyle-related behaviors, such as physical activity, screen time, and sleep, have been examined in isolation, with each behavior considered separately in relation to its effects on health outcomes [8]. However, in 2020, the Canadian Society for Exercise Physiology released the Canadian 24 h Movement Guidelines for Adults and Older Adults, introducing a more integrated approach to the study of these behaviors [8]. The guidelines promote the concept that “the whole day matters”, emphasizing the importance of assessing both the combined (joint) and individual (independent) effects of physical activity, screen time, and sleep on various health outcomes [8].

Most studies investigating the association between 24 h movement behaviors (physical activity, screen time, and sleep) and NCDs have been conducted in high-income countries [9,10]. However, this relationship remains underexplored in low- and middle-income countries such as Brazil, which faces significant health inequalities, including unequal distribution of resources, shortage of healthcare professionals, and insufficient infrastructure [11]. Therefore, studies with nationally representative samples are needed to examine the association between 24 h movement behaviors and NCDs in the Brazilian context. Such investigations may serve as a cost-effective epidemiological strategy to reduce the burden of NCDs, particularly within the Brazilian Unified Health System (*Sistema Único de Saúde*—SUS), the country’s publicly funded health system, whose primary goal is to ensure universal access to comprehensive healthcare. Article 2 of Law No. 8.080/1990 establishes that ‘Health is a fundamental right of the human being, and it is the duty of the State to provide the necessary conditions for its full exercise’ [12].

In this context, the objective of the present study was to examine the association between adherence to the 24 h movement guidelines (physical activity, screen time, and sleep) and the diagnosis of NCDs.

## 2. Materials and Methods

### 2.1. Design

This cross-sectional study used data from the “MOTIVA-SUS: Epidemiological Study on Motivational Determinants for Physical Activity Practice among Users of the Health Academy Program”. The Health Academy Program is a nationwide, ongoing health promotion initiative in Brazil that involves participation and funding from municipal, state, and federal governments. It is currently the most comprehensive health promotion program in the country. The program encompasses multiple components, including physical activity and body movement practices, healthcare and healthy lifestyle promotion, healthy eating, integrative and complementary practices, artistic and cultural activities, health education, planning and management, and community mobilization [13]. This study was conducted in accordance with the ethical principles of the Declaration of Helsinki and was approved by the Human Research Ethics Committee of the Federal University of Santa Catarina (Protocol number: 5.040.451). All participants provided informed consent prior to participation in this study.

Data collection was carried out through telephone interviews due to the adverse epidemiological context of the COVID-19 pandemic. The interviews took place between February and August 2022. All participants received detailed information regarding the informed consent form, which outlined the objectives and procedures of this study. Only those who agreed to participate were included in this study.

Participants who had any condition that prevented communication via telephone were excluded from the sample. Eligible individuals who could not be contacted after at least four phone call attempts, conducted on different days and at different times, including one on a weekend and another in the evening, were classified as refusals or sample losses. Additionally, eligible individuals who declined to participate in this study after being informed about its objectives were also categorized as refusals.

All analyses accounted for the study design and statistical power. A post hoc power analysis was conducted considering a type I error (α = 0.05) and type II error (β = 0.80) to estimate the minimum sample size required to detect associations between dependent and independent variables, based on the study design and planned binary and multinomial logistic regression models. A minimum of 296 adults and older adults was required for all analyses. Power calculations were performed using G*Power^®^ software, version 3.1.9.2 (University of Düsseldorf, Düsseldorf, Germany).

### 2.2. Dependent Variable (Outcome)

Information on NCDs was obtained through self-reported physician diagnoses using the following questions: “Has a doctor or healthcare professional ever told you that you have hypertension/high blood pressure?”; “Has a doctor or healthcare professional ever told you that you have diabetes/high blood sugar?”; “Has a doctor or healthcare professional ever told you that you have high cholesterol?”; and “Has a doctor or healthcare professional ever told you that you have heart disease or cardiovascular disease?”. Responses were dichotomous (No, Yes). Additionally, participants were asked whether they used specific medications for each of the listed NCDs. Only individuals who reported both a diagnosis and the use of medications for a given condition were classified as “Yes” for the respective NCD; all others were classified as “No”.

To construct the NCD presence variable, the number of diagnosed conditions reported by each participant (hypertension, diabetes, hypercholesterolemia, and cardiovascular diseases) was summed and transformed into an ordinal scale (0 = no NCD; 1 = one NCD; 2 = two NCDs; 3 = three or more NCDs). However, due to the small number of individuals who reported no NCD (n = 12; 1.1%), the categories “0” and “1” were combined for analytical purposes.

### 2.3. Independent Variable (Exposure)

Data on PA, screen time, and sleep duration and quality were collected using questions adapted from the validated questionnaire employed by the Brazilian Surveillance System for Risk and Protective Factors for Chronic Diseases by Telephone Survey (VIGITEL) [14].

Leisure time PA was assessed through the following questions: “On how many days per week do you usually engage in physical exercise or sports?”; and “On the days when you engage in exercise or sports, how long does the activity usually last?”. In this study, the following activities were classified as moderate-intensity PA: walking, treadmill walking, weight training, water aerobics, general gymnastics, swimming, martial arts, cycling, and volleyball. The following were considered vigorous-intensity PA: running, treadmill running, aerobic gymnastics, soccer, basketball, and tennis [14]. Participants who reported engaging in <150 min per week of moderate-intensity PA and/or <75 min per week of vigorous-intensity PA were classified as not meeting the recommendations. Conversely, individuals who performed ≥150 min of moderate-intensity PA and/or ≥75 min of vigorous-intensity PA per week were considered as meeting the recommendations [8,14,15].

Screen time was estimated based on responses to the following questions: “On a typical week, how many hours per day do you spend sitting and watching television from Monday to Friday?”; “On a typical week, how many hours per day do you spend sitting and watching television on Saturdays and Sundays?”; “On a typical week, on average, how many hours per day do you spend sitting in front of a computer, tablet, or smartphone from Monday to Friday?”; and “On a typical week, on average, how many hours per day do you spend sitting in front of a computer, tablet, or smartphone on Saturdays and Sundays?”. To estimate total daily screen time, a weighted average was calculated using the following formula: ((5 × weekday minutes) + (2 × weekend minutes))/7. Participants who reported ≥8 h per day of screen time were classified as not meeting the recommendation, while those with <8 h per day were classified as meeting the recommendation [8].

Sleep duration was estimated based on participants’ responses to the following questions: “At what time do you usually go to sleep on weekdays (Monday to Friday)?”; “At what time do you usually wake up on weekdays?”; “At what time do you usually go to sleep on weekends (Saturday and Sunday)?”; and “At what time do you usually wake up on weekends?”. Sleep quality was assessed through the question: “How would you rate the quality of your sleep?”, and responses were categorized as either good or poor. To estimate average sleep duration, the following weighted formula was applied: ((5 × weekday sleep minutes) + (2 × weekend sleep minutes))/7. Participants were then classified according to current sleep recommendations based on age and self-reported sleep quality [8]. For adults aged 18 to 64 years, individuals were considered to meet the recommendation if they reported 7 to 9 h of good-quality sleep per night. Those who reported less than 7 h, more than 9 h, or 7 to 9 h of poor-quality sleep were classified as not meeting the recommendation. For older adults aged 65 years and above, those who reported 7 to 8 h of good-quality sleep were classified as meeting the recommendation, while individuals with less than 7 h, more than 8 h, or 7 to 8 h of poor-quality sleep were considered as not meeting the recommendation.

Based on the classification of each movement behavior according to current guidelines [8], a variable was created to capture the simultaneity of adherence to the 24 h movement behaviors. This variable included the following categories: “meeting all three recommendations”; “meeting only the PA recommendation”; “meeting only the screen time recommendation”; “meeting only the sleep recommendation”; “meeting both the PA and screen time recommendations”; “meeting both the PA and sleep recommendations”; “meeting both the screen time and sleep recommendations”; and “not meeting any of the recommendations”.

### 2.4. Covariates

The covariates included in this study were as follows: sex (male, and female); age in years (19 to 59 years, and ≥60 years); educational level (≤8 years of schooling, 9 to 11 years, and ≥12 years); and marital status (married and single/divorced/widowed). Body mass index (BMI) was calculated by dividing body mass in kilograms by height in meters squared (body mass/height^2^) [16]. BMI was categorized as “normal weight” and “overweight + obesity”, in accordance with WHO recommendations [16].

### 2.5. Statistical Analysis

Descriptive statistics were used to characterize the sample and are presented as absolute and relative frequencies. To examine the association between isolated and combined 24 h movement behaviors (PA, screen time, and sleep) and the presence of NCDs (hypertension, diabetes, hypercholesterolemia, and cardiovascular diseases), Wald tests and binary logistic regression analyses were performed. Results are presented as odds ratios (OR) with 95% confidence intervals (95% CI).

The binary logistic regression models were adjusted for sex, age, educational level, marital status, and BMI. The final adjusted model was built using a forward selection procedure, considering all covariates regardless of their *p*-values in the unadjusted analysis. Variables with a *p*-value < 0.20 in the final adjusted model were retained in the final model. The goodness of fit of the binary logistic regression models was assessed using the area under the ROC curve (AUC), the Hosmer–Lemeshow test, and Nagelkerke’s R^2^. In addition, the Variance Inflation Factor (VIF) was used to check for multicollinearity among the independent variables (Appendix A).

Additionally, multinomial logistic regression and Wald tests were used to investigate the association between the simultaneous presence of multiple NCDs (hypertension, diabetes, hypercholesterolemia, and cardiovascular diseases) within the same individual (dependent variable) and 24 h movement behaviors (PA, screen time, and sleep), considered both individually and jointly. Results are expressed as ORs with 95% CIs, with the reference category being the group with no/one NCD. The multinomial logistic regression models were adjusted for sex, age, educational level, marital status, and BMI. The final adjusted model was built using a forward selection procedure, considering all covariates regardless of their *p*-values in the unadjusted analysis. Variables with a *p*-value < 0.20 in the final adjusted model were retained in the final model. The goodness-of-fit of the multinomial logistic regression models was assessed using Nagelkerke’s R^2^, and the Variance Inflation Factor (VIF) was used to detect potential multicollinearity issues among the independent variables (Appendix A).

All descriptive and regression analyses were performed using the Statistical Package for the Social Sciences (IBM SPSS Statistics, Chicago, IL, USA), version 23.0. A significance level of 5% was adopted, and *p*-values < 0.05 were considered statistically significant. The correction of *p*-values for multiple comparisons was performed using the Bonferroni–Hochberg method, with the aid of Stata software (StataCorp LP, College Station, TX, USA), version 16.0.

## 3. Results

Regarding sample characteristics, the majority of participants were female (n = 1126; 92.9%), aged between 19 and 59 years (n = 897; 74.1%), married (n = 745; 61.5%), classified as overweight or obese (n = 809; 73.0%), and had 12 or more years of schooling (n = 720; 59.5%). With respect to health conditions, 737 individuals (60.9%) reported having hypertension, 983 (81.4%) had diabetes, 763 (64.1%) reported hypercholesterolemia, and 1125 (92.9%) indicated having some form of cardiovascular disease. In addition, 829 participants (69.9%) presented three or more of these conditions simultaneously (hypertension, diabetes, hypercholesterolemia, and cardiovascular disease). In terms of 24 h movement behaviors, 606 participants (50.0%) met the PA recommendations, 553 (46.4%) met the sleep recommendations, and 577 (60.4%) met the screen time recommendations. Only 150 individuals (15.9%) adhered to all three 24 h movement guidelines simultaneously, while 109 participants (11.5%) did not meet any of the recommendations (Table 1).

Participants who did not meet any of the 24 h movement behavior recommendations had higher odds of hypertension (Table 2), diabetes (Table 3), and hypercholesterolemia (Table 4).

A statistically significant association was also observed among individuals who met only the PA recommendation, as they had lower odds of cardiovascular disease in both the crude analysis and the adjusted model. Likewise, participants who simultaneously met the PA and sleep recommendations had lower odds of cardiovascular disease in both the crude analysis and the adjusted model (Table 5).

Participants who did not meet any of the 24 h movement behavior recommendations also had higher odds of having two NCDs or three or more NCDs in the crude analysis. These associations remained statistically significant in the adjusted model, with an OR of 1.29 (95% CI: 1.09–2.91; *p* = 0.04) for two NCDs and an OR of 1.40 (95% CI: 1.11–2.13; *p* = 0.02) for three or more NCDs (Table 6).

## 4. Discussion

The main findings of this study indicate that adherence to movement behavior guidelines, even partial adherence (meeting only the PA recommendation or PA combined with sleep), was associated with lower odds of cardiovascular disease. In addition, not meeting the PA recommendation was associated with higher odds of hypercholesterolemia. Furthermore, not adhering to any of the three movement behavior recommendations (PA, screen time, and sleep) was associated with increased odds of hypertension, diabetes, and having two or more noncommunicable diseases.

In the present study, not meeting any of the 24 h movement behavior recommendations was associated with higher odds of having two or more NCDs. Evidence from adult populations suggests that noncompliance with these guidelines is linked to adverse health outcomes such as diabetes, hypertension, coronary heart disease, and an increased risk of cardiovascular conditions [17,18,19]. In this context, insufficient PA has been associated with unfavorable physiological responses, including dysregulation in the production of pro-inflammatory cytokines such as tumor necrosis factor-alpha (TNF-α), interleukin-6 (IL-6), interleukin-1 beta (IL-1β), and C-reactive protein (CRP), all of which are biomarkers related to NCDs [20,21]. Similarly, excessive screen time may contribute to the development of atherosclerosis, vascular stiffness, elevated levels of pro-inflammatory cytokines, and increased insulin resistance, which are also considered important risk factors for NCDs [22]. Additionally, sleep plays a critical role in regulating hormones that maintain homeostatic balance in the body [23]. Irregular sleep patterns can disrupt circadian rhythms, resulting in increased production of insulin, TNF-α, and IL-6 [24].

Furthermore, not meeting the recommendations for all three movement behaviors was associated with higher odds of hypertension in the present study. Simultaneous noncompliance with PA, sleep, and screen time guidelines may elevate the risk of hypertension, as insufficient PA impairs cardiac output regulation, influences sympathetic nervous system activity, and increases peripheral vascular resistance, all of which are key mechanisms involved in blood pressure elevation [25]. Excessive screen time may also lead to adverse hemodynamic changes, such as vascular dysfunction and reduced shear stress in blood vessels, further contributing to elevated blood pressure [22]. Moreover, both insufficient and excessive sleep duration can alter sympathetic nervous system activity, increase cortisol secretion, and dysregulate the hypothalamic–pituitary–adrenal axis, thereby favoring the development of hypertension [24].

Not meeting the 24 h movement behavior guidelines was associated with higher odds of diabetes in the participants of this study. Similar findings have been reported in the literature. A study conducted with adults in Chile found that nonadherence to movement guidelines was associated with increased odds of type 2 diabetes diagnosis [19]. Other studies involving adult populations have also identified associations between 24 h movement behaviors and diabetes [17,18]. Furthermore, research has shown that both short and long sleep durations are associated with the development of type 2 diabetes, likely due to circadian rhythm misalignment, which can impair glycemic control [23]. Similarly, physical inactivity and excessive screen time have been linked to reduced glucose uptake and utilization by skeletal muscle, decreased insulin sensitivity, and overall impaired glycemic regulation [21,26].

The present study showed that individuals who met only the PA recommendations, or both PA and sleep recommendations, had lower odds of cardiovascular disease. These findings are consistent with previous studies conducted among adults, which have demonstrated that meeting PA guidelines reduces the risk of cardiovascular conditions [10,27,28]. A possible explanation lies in the fact that regular PA stimulates the release of myokines, which regulate inflammatory markers such as TNF-α, IL-6, IL-1β, and CRP, all of which are associated with cardiovascular disease [20]. Similarly, adequate sleep contributes to the regulation of pro-inflammatory cytokine production, including TNF-α, IL-6, and CRP [29].

Participants in the present study who did not meet the PA recommendation had higher odds of hypercholesterolemia. Previous studies conducted among adults have also identified an association between PA and hypercholesterolemia [30,31]. This relationship can be explained by the regulatory role that physical activity plays on lipoprotein lipase, an enzyme responsible for the removal of circulating triglycerides, which contributes to the reduction in total cholesterol levels [32]. In addition, insufficient PA impairs fatty acid oxidation and promotes an increase in the production of low-density lipoprotein (LDL) cholesterol [33].

This study presents some limitations. Among them, data collection was conducted through a telephone survey, a feasible method for epidemiological research involving large population groups; however, the responses may be influenced by recall bias [34]. Accordingly, the diagnosis of NCDs was obtained indirectly through self-report. Furthermore, the analysis included only the four NCDs most frequently addressed in the literature (hypertension, diabetes, dyslipidemia, and cardiovascular diseases) [10,18,19,30]. It is possible that including other NCDs in the analysis might have influenced the results. In addition, the absence of data on smoking, alcohol consumption, and income represents a limitation, as these variables may act as confounding factors, influencing the estimated effects between 24 h movement behaviors and health outcomes, which may result in either overestimation or underestimation of the observed associations. Similarly, 24 h movement behaviors were assessed via self-report, which may be subject to recall bias and result in over- or underestimation of the time spent in each behavior, potentially affecting the observed associations. Although the objective of this study was not to compare PA levels, screen time, and sleep duration with data from the pre-COVID-19 period, previous studies have shown that, during the pandemic, there was a decrease in PA [35], an increase in screen time [36], and changes in sleep patterns [37]. Therefore, the generalizability of the findings is limited. Another limitation of this study refers to the imbalance in the sample composition, which is predominantly female (92.9%). This considerable representation reduces the possibility of generalizing the findings to the overall adult Brazilian population, especially with regard to potential sex differences in the associations investigated. Additionally, due to the cross-sectional nature of this study, temporal relationships between exposure and outcomes cannot be established, and the possibility of reverse causality must be acknowledged, that is, individuals with NCDs may have modified their PA, screen time, or sleep patterns after the onset of their condition. Despite these limitations, this study has notable strengths. The questions used were based on validated instruments for the target population. The use of a complex sampling design [38] allowed for the assignment of sampling weights and improved representativeness. Moreover, this study included participants from all five geographic regions of Brazil.

## 5. Conclusions

Participants of the Health Academy Program in Brazil who did not meet any of the 24 h movement behavior recommendations had higher odds of hypertension, diabetes, and multimorbidity. Additionally, noncompliance with the PA guideline among these participants was associated with increased odds of hypercholesterolemia. In contrast, those who adhered to the PA recommendation had lower odds of cardiovascular disease. These findings suggest that 24 h movement behaviors may represent a low-cost epidemiological strategy for the prevention and control of NCDs within the Brazilian Unified Health System. The national health system could consider developing awareness campaigns emphasizing the importance of meeting PA recommendations, reducing screen time, and promoting adequate sleep duration and quality. In this regard, it is suggested that the 24 h movement guidelines be integrated into primary care protocols through simple screening and counseling tools, such as questionnaires. Moreover, these recommendations may be incorporated into primary health care interventions to help prevent NCDs. Likewise, in clinical settings, health professionals could encourage discussions not only about PA, but also about other movement behaviors and their implications for health. In addition, the training of SUS professionals to deliver educational approaches that integrate physical activity, sleep, and screen time is recommended. Leveraging existing programs, such as the Health Academy Program, may support the implementation of multicomponent and low-cost interventions. In this context, understanding the adverse effects of inadequate movement behaviors, whether individually or in combination, may help reduce healthcare costs and mortality associated with NCDs. Finally, future studies with longitudinal designs or experimental interventions may contribute to a better understanding of the effects of 24 h movement behaviors on NCDs.

## Figures and Tables

**Table 1 healthcare-13-02016-t001:** Distribution of sample characteristics among adult and older adult participants of the Health Academy Program in Brazil.

Variable	n	% (95% CI)
**Sex**		
Female	1126	92.9% (91.39–94.41)
Male	86	7.1% (1.66–12.54)
**Age**		
19–59 years	897	74.1% (71.23–76.97)
≥60 years	314	25.9% (21.07–30.73)
**Marital status**		
Married	745	61.5% (58.01–64.99)
Single/Divorced/Widowed	467	38.5% (34.09–42.91)
**BMI**		
Normal weight	299	27.0% (21.96–32.04)
Overweight/Obesity	809	73.0% (69.94–76.06)
**Educational level**		
≤8 years of schooling	233	19.3% (13.98–24.08)
9–11 years of schooling	257	21.2% (16.20–26.20)
≥12 years of schooling	720	59.5% (55.92–63.08)
**Hypertension**		
Yes	737	60.9% (57.37–64.43)
No	474	39.1% (34.70–43.50)
**Diabetes**		
Yes	983	81.4% (78.97–83.83)
No	224	18.6% (13.52–23.68)
**Hypercholesterolemia**		
Yes	763	64.1% (60.70–67.50)
No	427	35.9% (31.35–40.45)
**Cardiovascular disease**		
Yes	1125	92.9% (91.40–94.40)
No	86	7.1% (1.66–12.54)
**Presence of NCDs**		
0/1 NCD	108	9.1% (3.66–14.54)
2 NCDs	249	21.0% (15.95–26.05)
3 or more NCDs	829	69.9% (66.78–73.02)
**PA**		
Meets the recommendation	606	50.0 (46.02–53.98)
Does not meet the recommendation	605	50.0 (46.02–53.98)
**Sleep**		
Meets the recommendation	553	46.4% (42.22–50.58)
Does not meet the recommendation	639	53.6% (49.72–57.48)
**Screen Time**		
Meets the recommendation	577	60.4% (56.10–64.70)
Does not meet the recommendation	378	39.6% (35.09–44.11)
**Adherence to 24 h movement behavior recommendations**		
Meets all three recommendations	150	15.9% (13.00–19.00)
Meets only the PA recommendation	90	9.5% (6.60–12.40)
Meets only the screen time recommendation	153	16.2% (13.30–19.10)
Meets only the sleep recommendation	124	13.1% (10.30–16.0)
Meets only the PA and screen time recommendations	89	9.4% (6.50–12.30)
Meets only the PA and sleep recommendations	86	9.1% (6.20–12.00)
Meets only the screen time and sleep recommendations	143	15.1% (12.30–18.0)
Does not meet any of the recommendations	109	11.5% (8.70–14.30)

Note. NCD: noncommunicable diseases; PA: physical activity; CI: confidence interval; n: sample size.

**Table 2 healthcare-13-02016-t002:** Association between 24 h movement behaviors and hypertension among adult and older adult participants of the Health Academy Program in Brazil.

	Hypertension						
Variable	OR	Crude (95% CI)	*p*	OR	Adjusted * (95% CI)	*p*	*p* Bonferroni–Hochberg
**PA**							
Meets the recommendation	1			1			
Does not meet the recommendation	1.05	(0.83–1.32)	0.67	0.94	(0.73–1.22)	0.66	0.77
**Sleep**							
Meets the recommendation	1			1			
Does not meet the recommendation	0.85	(0.67–1.08)	0.19	0.95	(0.74–1.23)	0.68	0.82
**Screen Time**							
Meets the recommendation	1			1			
Does not meet the recommendation	1.11	(0.84–1.45)	0.44	0.92	(0.68–1.25)	0.61	0.88
**Adherence to 24 h movement behavior recommendations**							
Meets all three recommendations	1			1			
Meets only the PA recommendation	0.90	(0.52–1.56)	0.72	1.00	(0.56–173)	0.99	0.99
Meets only the screen time recommendation	0.85	(0.53–1.37)	0.52	1.12	(10.61–2.04)	0.71	0.81
Meets only the sleep recommendation	1.05	(0.63–1.74)	0.85	0.93	(0.57–1.55)	0.80	0.82
Meets only the PA and screen time recommendations	1.05	(0.51–1.54)	0.68	1.21	(0.70–2.09)	0.49	0.52
Meets only the PA and sleep recommendations	0.89	(0.53–1.63)	0.80	1.15	(0.63–2.13)	0.63	0.67
Meets only the screen time and sleep recommendations	0.93	(0.52–1.73)	0.50	1.01	(0.49–1.39)	0.48	0.53
Does not meet any of the recommendations	1.71	(1.42–2.34)	<0.01 †	1.35	(1.15–1.77)	<0.01 †	<0.01 †

Note. PA: physical activity; CI: confidence interval; OR: odds ratio; †: *p* < 0.05; * analysis adjusted for sex, age, educational level, body mass index, and marital status.

**Table 3 healthcare-13-02016-t003:** Association between 24 h movement behaviors and diabetes among adult and older adult participants of the Health Academy Program in Brazil.

	Diabetes						
Variable	OR	Crude (95% CI)	*p*	OR	Adjusted * (95% CI)	*p*	*p* Bonferroni–Hochberg
**PA**							
Meets the recommendation	1			1			
Does not meet the recommendation	1.33	(0.99–1.78)	0.55	0.82	(0.60–1.13)	0.23	0.27
**Sleep**							
Meets the recommendation	1			1			
Does not meet the recommendation	1.01	(0.75–1.33)	0.94	0.88	(0.64–1.21)	0.46	0.87
**Screen Time**							
Meets the recommendation	1			1			
Does not meet the recommendation	1.12	(0.79–1.58)	0.55	0.92	(0.63–1.34)	0.68	0.98
**Adherence to 24 h movement behavior recommendations**							
Meets all three recommendations	1			1			
Meets only the PA recommendation	0.79	(0.40–1.55)	0.49	1.19	(0.57–2.49)	0.06	0.07
Meets only the screen time recommendation	0.70	(0.35–1.43)	0.33	0.84	(0.40–1.75)	0.06	0.07
Meets only the sleep recommendation	0.65	(0.35–1.21)	0.17	0.75	(0.40–1.41)	0.88	0.89
Meets only the PA and screen time recommendations	0.77	(0.40–1.48)	0.43	0.95	(0.48–1.87)	0.08	0.38
Meets only the PA and sleep recommendations	0.76	(0.37–1.54)	0.45	1.10	(0.51–2.30)	0.07	0.35
Meets only the screen time and sleep recommendations	1.72	(0.35–1.48)	0.38	1.03	(0.42–1.85)	0.08	0.09
Does not meet any of the recommendations	1.02	(1.00–1.78)	0.02 †	1.07	(1.03–2.01)	<0.01 †	<0.01 †

Note. PA: physical activity; CI: confidence interval; OR: odds ratio; †: *p* < 0.05; * analysis adjusted for sex, age, educational level, body mass index, and marital status.

**Table 4 healthcare-13-02016-t004:** Association between 24 h movement behaviors and hypercholesterolemia among adult and older adult participants of the Health Academy Program in Brazil.

	Hypercholesterolemia						
Variable	OR	Crude (95% CI)	*p*	OR	Adjusted * (95% CI)	*p*	*p* Bonferroni–Hochberg
**PA**							
Meets the recommendation	1			1			
Does not meet the recommendation	1.08	(1.06–1.95)	0.03 †	1.37	(1.06–1.76)	<0.01 †	<0.01 †
**Sleep**							
Meets the recommendation	1			1			
Does not meet the recommendation	1.13	(0.89–1.44)	0.30	1.15	(0.89–1.49)	0.26	0.28
**Screen Time**							
Meets the recommendation	1			1			
Does not meet the recommendation	1.29	(0.98–1.70)	0.06	1.20	(0.89–1.62)	0.21	0.29
**Adherence to 24 h movement behavior recommendations**							
Meets all three recommendations	1			1			
Meets only the PA recommendation	0.96	(0.56–1.64)	0.88	1.12	(0.63–2.00)	0.68	0.74
Meets only the screen time recommendation	1.11	(0.62–1.19)	0.71	1.34	(0.73–2.45)	0.34	0.34
Meets only the sleep recommendation	1.04	(0.63–1.70)	0.87	1.08	(0.64–1.80)	0.76	0.82
Meets only the PA and screen time recommendations	1.08	(0.64–1.81)	0.76	1.25	(0.72–2.16)	0.41	0.46
Meets only the PA and sleep recommendations	0.60	(0.35–1.04)	0.72	0.63	(0.35–1.14)	0.13	0.16
Meets only the screen time and sleep recommendations	0.58	(0.33–1.01)	0.56	0.57	(0.31–1.04	0.69	0.78
Does not meet any of the recommendations	0.94	(0.57–1.54)	0.80	0.87	(0.52–1.47)	0.61	0.62

Note. PA: physical activity; CI: confidence interval; OR: odds ratio; †: *p* < 0.05; * analysis adjusted for sex, age, educational level, body mass index, and marital status.

**Table 5 healthcare-13-02016-t005:** Association between 24 h movement behaviors and cardiovascular diseases among adult and older adult participants of the Health Academy Program in Brazil.

	Cardiovascular Disease						
Variable	OR	Crude (95% CI)	*p*	OR	Adjusted * (95% CI)	*p*	*p* Bonferroni–Hochberg
**PA**							
Meets the recommendation	1			1			
Does not meet the recommendation	1.05	(0.67–1.63)	0.83	1.15	(0.72–1.84)	0.55	0.57
**Sleep**							
Meets the recommendation	1			1			
Does not meet the recommendation	1.47	(0.93–2.31)	0.09	0.75	(0.46–1.22)	0.24	0.25
**Screen Time**							
Meets the recommendation	1			1			
Does not meet the recommendation	1.68	(0.98–2.80)	0.06	0.71	(0.41–1.25)	0.24	0.26
**Adherence to 24 h movement behavior recommendations**							
Meets all three recommendations	1			1			
Meets only the PA recommendation	0.31	(0.11–0.84)	0.02 †	0.32	(0.11–0.85)	0.02 †	0.02 †
Meets only the screen time recommendation	1.21	(0.29–4.99)	0.78	0.84	(0.41–7.61)	0.43	0.48
Meets only the sleep recommendation	0.67	(0.23–1.93)	0.46	1.03	(0.28–2.51)	0.76	0.83
Meets only the PA and screen time recommendations	0.82	(0.25–2.62)	0.74	0.95	(0.31–3.37)	0.95	0.98
Meets only the PA and sleep recommendations	0.33	(0.11–0.94)	0.03 †	0.32	(0.11–0.90)	0.03 †	0.04 †
Meets only the screen time and sleep recommendations	0.55	0.17–1.79)	0.36	1.69	(0.19–2.43)	0.56	0.59
Does not meet any of the recommendations	0.62	(0.21–1.80)	0.38	1.84	(0.75–1.92)	0.18	0.21

Note. PA: physical activity; CI: confidence interval; OR: odds ratio; †: *p* < 0.05; * analysis adjusted for sex, age, educational level, body mass index, and marital status.

**Table 6 healthcare-13-02016-t006:** Association between 24 h movement behaviors and the simultaneous presence of noncommunicable diseases (hypertension, diabetes, hypercholesterolemia, and cardiovascular diseases) among adult and older adult participants of the Health Academy Program in Brazil (reference category: 0 or 1 noncommunicable diseases).

	Two Noncommunicable Diseases							≥Three Noncommunicable Diseases						
Variable	OR	Crude (95% CI)	*p*	OR	Adjusted * (95% CI)	*p*	p Bonferroni–Hochberg	OR	Crude (95% CI)	*p*	OR	Adjusted * (95% CI)	*p*	p Bonferroni–Hochberg
**PA**														
Meets the recommendation	1			1				1			1			
Does not meet the recommendation	1.37	(0.87–2.16)	0.17	1.25	(0.78–2.01)	0.34	0.37	1.12	(0.75–1.68)	0.56	1.15	(0.75–1.76)	0.51	0.71
**Sleep**														
Meets the recommendation	1			1				1			1			
Does not meet the recommendation	0.60	(0.38–1.96)	0.36	0.62	(0.38–1.02)	0.06	0.08	0.64	(0.42–1.97)	0.39	0.71	(0.45–1.11)	0.14	0.27
**Screen Time**														
Meets the recommendation	1			1				1			1			
Does not meet the recommendation	0.65	(0.38–1.14)	0.13	0.68	(0.38–1.21)	0.19	0.21	1.55	(0.34–1.91)	0.20	0.68	(0.41–1.17)	0.17	0.28
**Adherence to 24 h movement behavior recommendations**														
Meets all three recommendations	1			1				1			1			
Meets only the PA recommendation	1.16	(0.28–4.67)	0.82	1.22	(0.30–5.01)	0.77	0.83	1.16	(0.29–4.67)	0.82	0.48	(0.25–0.92)	0.28	0.31
Meets only the screen time recommendation	0.55	(0.18–1.68)	0.29	0.56	(0.18–1.74)	0.32	0.38	0.55	(0.18–1.68)	0.29	0.40	(0.14–1.13)	0.84	0.85
Meets only the sleep recommendation	0.92	(0.26–3.25)	0.90	0.97	(0.27–3.47)	0.97	0.98	0.92	(0.26–3.25)	0.90	1.14	(0.30–4.35)	0.26	0.26
Meets only the PA and screen time recommendations	0.40	(0.12–1.29)	0.12	0.47	(0.14–1.59)	0.22	0.27	0.40	(0.24–1.29)	0.12	1.04	(0.31–3.40)	0.94	0.96
Meets only the PA and sleep recommendations	0.44	(0.13–1.46)	0.18	0.45	(0.13–1.58)	0.21	0.32	0.44	(0.13–1.46)	0.18	0.47	(0.15–1.44)	0.19	0.25
Meets only the screen time and sleep recommendations	0.51	(0.16–1.16)	0.25	0.52	(0.16–1.67)	0.27	0.28	0.51	(0.16–1.62)	0.25	0.39	(0.20–1.72)	0.11	0.14
Does not meet any of the recommendations	1.28	(1.08–2.88)	0.02 †	1.29	(1.09–2.91)	0.04 †	0.04 †	1.28	(1.10–2.76)	<0.01 †	1.40	(1.11–2.13)	0.02 †	0.03 †

Note: PA: physical activity; CI: confidence interval; OR: odds ratio; †: *p* < 0,05; * analysis adjusted for sex, age, educational level, body mass index, and marital status.

## Data Availability

The raw data supporting the conclusions of this study will be made available by the corresponding author on request.

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
