# Peer review of "Association Between 24-Hour Movement Behaviors and Noncommunicable Chronic Diseases Among Adult and Older Adult Users of the Brazilian Community Health Promotion Program"

_healthcare, 2025, doi:10.3390/healthcare13162016_

Round 1

Reviewer 1 Report

Comments and Suggestions for Authors

This study investigated the link between 24-hour movement behaviors (physical activity, screen time, and sleep) and noncommunicable chronic diseases (NCDs) in Brazilian adults. It found that not adhering to recommended movement guidelines, particularly for physical activity, increased the odds of NCDs like hypertension, diabetes, and hypercholesterolemia. Conversely, meeting physical activity guidelines was associated with lower odds of cardiovascular disease. The findings emphasize the need for an integrated approach to lifestyle for NCD prevention. Overall, the paper is nicely written. However, there are few major methodological issues.

1. The study population is not very representative of Brazilian adults, with majority being women. Authors should at least acknowledge that these findings might not be generalizable. Was the sex proportion similarly imbalanced in the Health Academy Program as well? What are the participants characteristics differences between the overall Health Academy Program vs. people who enrolled in this study.

2. The study was conducted during covid-19 pandemic, when everyone's routine changed. Have the authors tried to compare if their population statistics, in terms of PA, screen time, and sleep time, similar to the previous years or not? Screen time could be artificially elevated due to the pandemic, which makes the results even less generalizable. 

3. The statistical methods have some major conflicts:  "All covariates were included in the adjusted model regardless of their p-values in the unadjusted analysis. A forward selection procedure was applied to the adjusted model, whereby each covariate was entered stepwise. Covariates with a Wald test p-value below 0.20 were retained in the final adjusted model. " What is the final method applied? Forward selection or adjusting all covariates?

4. Multiple comparison issue is the major flaw of the current analyses. There are so many associations/hypotheses the authors tested. And only a few are significant. However, in the abstract, none of these is mentioned. Multiple comparison should be addressed and bonferroni-corrected p values or false discovery rate should be used instead of the common p<0.05.

Author Response

Reviewer Comments:

Reviewer #1: This study investigated the link between 24-hour movement behaviors (physical activity, screen time, and sleep) and noncommunicable chronic diseases (NCDs) in Brazilian adults. It found that not adhering to recommended movement guidelines, particularly for physical activity, increased the odds of NCDs like hypertension, diabetes, and hypercholesterolemia. Conversely, meeting physical activity guidelines was associated with lower odds of cardiovascular disease. The findings emphasize the need for an integrated approach to lifestyle for NCD prevention. Overall, the paper is nicely written. However, there are few major methodological issues.

Authors: We acknowledge the importance of the methodological considerations and are aware of the potential limitations of the study. We clarify that the methodological decisions were made based on the characteristics of the sample and the available analytical possibilities. We remain open to reviewing, further exploring, or clarifying the points raised.

Reviewer #1: The study population is not very representative of Brazilian adults, with majority being women. Authors should at least acknowledge that these findings might not be generalizable. Was the sex proportion similarly imbalanced in the Health Academy Program as well? What are the participants characteristics differences between the overall Health Academy Program vs. people who enrolled in this study.

Authors: We appreciate your suggestions and agree with the comments. Indeed, we acknowledge that the sample in the present study is predominantly composed of women (92.9%), which limits the generalizability of the findings to the overall adult Brazilian population. This limitation has already been mentioned in the discussion section and will be more clearly emphasized, as suggested. Regarding the sex distribution, we note that the imbalance observed in the sample is consistent with the general profile of users of the Health Academy Program as identified in previous surveys conducted by the Brazilian Ministry of Health and in studies on the program (Costa et al., 2019; Caldeira et al., 2022; Gonçalves et al., 2024).

In the current version of the manuscript, the following revision has been made:

Another limitation of the study refers to the imbalance in the sample composition, which is predominantly female (92.9%). This considerable representation reduces the possibility of generalizing the findings to the overall adult Brazilian population, especially with regard to potential sex differences in the associations investigated”.

References:

Costa BVL, Menezes MC, Oliveira CDL, Mingoti SA, Jaime PC, Caiaffa WT, Lopes ACS. Does access to healthy food vary according to socioeconomic status and to food store type? an ecologic study. BMC Public Health. 2019 Jun 18;19(1):775. doi: 10.1186/s12889-019-6975-y. PMID: 31215435; PMCID: PMC6582565.

Caldeira TCM, Soares MM, Silva LESD, Veiga IPA, Claro RM. Comportamentos de Risco e Proteção para Doenças Crônicas nas Capitais Brasileiras e no Distrito Federal, segundo a Pesquisa Nacional de Saúde e o Sistema de Vigilância Telefônica de Fatores de Risco e Proteção para Doenças Crônicas, 2019. Epidemiol Serv Saude. 2022;31(1):e2021367. doi: 10.1590/SS2237-9622202200009.

Gonçalves L, Zanlorenci S, Pelegrini A, Lima TR, Silva DAS. Individual and Joint Association between Cardiovascular Disease Risk Factors and Inadequate Lifestyle Behaviors in a Sample from Brazil. Arq Bras Cardiol. 2024 Oct 18;121(10):e20240149. Portuguese, English. doi: 10.36660/abc.20240149.

 Reviewer #1: The study was conducted during covid-19 pandemic, when everyone's routine changed. Have the authors tried to compare if their population statistics, in terms of PA, screen time, and sleep time, similar to the previous years or not? Screen time could be artificially elevated due to the pandemic, which makes the results even less generalizable. 

Authors: We appreciate your suggestions and agree with the comments. This study did not include a comparison of the data with the years preceding the COVID-19 pandemic. It is important to note that the COVID-19 pandemic led to significant changes in population routines, which may have affected movement behaviors, including increased screen time and reduced physical activity. Although our study did not conduct a direct comparison with pre-pandemic population data, we acknowledge that this limitation was taken into account when interpreting the results. Based on the reviewer’s suggestion, we will include in the Discussion section a critical reflection on how the pandemic context may have influenced the measures of physical activity, screen time, and sleep, thereby reinforcing the limitations regarding the generalizability of our findings. Indeed, data collection was conducted during the COVID-19 pandemic, which may have impacted 24-hour movement behavior patterns, particularly with respect to increased screen time and decreased physical activity. We recognize that this exceptional context may have influenced the results and limits the generalizability of the findings. In the current version of the manuscript, the following revision was made:

 “Similarly, 24-hour movement behaviors were assessed via self-report, which may be subject to recall bias and result in over- or underestimation of the time spent in each behavior, potentially affecting the observed associations. Although the objective of this study was not to compare physical activity levels, screen time, and sleep duration with data from the pre-COVID-19 period, previous studies have shown that during the pandemic there was a decrease in physical activity [35], an increase in screen time [36], and changes in sleep patterns [37]. Therefore, the generalizability of the findings is limited.

Reviewer #1: The statistical methods have some major conflicts:  "All covariates were included in the adjusted model regardless of their p-values in the unadjusted analysis. A forward selection procedure was applied to the adjusted model, whereby each covariate was entered stepwise. Covariates with a Wald test p-value below 0.20 were retained in the final adjusted model. " What is the final method applied? Forward selection or adjusting all covariates?

Authors: We appreciate your suggestions and agree with the comments. We have revised the statistical methods section to make it clearer for the reader. In the current version of the manuscript, the following revision was made:

The final adjusted model was built using a forward selection procedure, considering all covariates regardless of their p-values in the unadjusted analysis. Variables with a p-value < 0.20 in the final adjusted model were retained in the final model.”

Reviewer #1: Multiple comparison issue is the major flaw of the current analyses. There are so many associations/hypotheses the authors tested. And only a few are significant. However, in the abstract, none of these is mentioned. Multiple comparison should be addressed and bonferroni-corrected p values or false discovery rate should be used instead of the common p<0.05.

Authors: We appreciate the relevant observation regarding the control of multiple comparisons. We recognize the importance of accounting for the risk of Type I error when multiple associations are tested. In response to the suggestion, we chose to apply the Benjamini-Hochberg correction instead of the Bonferroni correction. This decision is based on the fact that the Benjamini-Hochberg approach is less conservative and more appropriate in contexts where hypotheses are theoretically grounded and there is interest in preserving statistical power (Amstrong, 2014; Howell, 2013). This method allows for control of the false discovery rate, offering a more suitable balance between sensitivity and Type I error control (Amstrong, 2014; Howell, 2013), especially when evaluating a moderate to high number of comparisons, as is the case in this study.

References:

Armstrong, R. A. (2014). When to use the Bonferroni correction. Ophthalmic and Physiological Optics, 34(5), 502–508. https://doi.org/10.1111/opo.12131

Howell, D. C. (2013). Multiple comparisons among treatment means. In D. C. Howell, Statistical methods for psychology (8th ed., pp. 369–410). Wadsworth Cengage Learning.

Reviewer 2 Report

Comments and Suggestions for Authors

Introduction
- The authors are encouraged to include more supporting evidence from other low- and middle-income countries (LMICs) to justify the focus on Brazil further.
- The sentence spanning Lines 68 to 75 cites reference number [12] twice unnecessarily. It is advisable to consolidate this into a single citation for clarity and conciseness.

Methods
- While the use of telephone interviews was appropriate given the constraints of the COVID-19 pandemic, it may have introduced potential information bias. This methodological limitation should be discussed more critically either in the Methods section or within the Discussion to acknowledge its impact on data validity.

Results
- There is significant repetition of numerical data already displayed in the tables. It would enhance the narrative to summarise the core findings rather than restating numbers verbatim.
- Considering the overwhelmingly female composition of the sample, it is recommended to perform subgroup or sensitivity analyses by sex to evaluate potential differences in the observed associations.

Discussion
- A more thorough discussion is needed regarding the gender imbalance in the sample and how this may limit the external validity or generalizability of the findings.
- The authors should also elaborate on the practical implications of their results, especially how these insights could inform public health strategies or interventions within the framework of Brazil's Unified Health System (SUS).

Conclusion
- The manuscript would be strengthened by suggesting directions for future studies, such as the inclusion of longitudinal research designs or experimental interventions to establish causality.

References
- In reference number 1, the source is written as "WORLD HEALTH ORGANIZATION" in all capital letters. Unless it is a style requirement of the target journal, it is advisable to use sentence case (i.e., "World Health Organization") for consistency with other reference entries and standard citation practices.

Author Response

Reviewer 2 

Reviewer #2: The authors are encouraged to include more supporting evidence from other low- and middle-income countries (LMICs) to justify the focus on Brazil further.

Authors: We appreciate your suggestions and agree with the comments. In the current version of the manuscript, the following revision was made:

“Low- and middle-income countries are experiencing a significant increase in the prevalence of type 2 diabetes [3] and multimorbidity [4], whereas the burden of noncommunicable diseases (NCDs) is decreasing in high-income countries; however, the same trend is not observed in low- and middle-income countries [5].”

Reviewer #2: The sentence spanning Lines 68 to 75 cites reference number [12] twice unnecessarily. It is advisable to consolidate this into a single citation for clarity and conciseness.

Authors: We appreciate your suggestions and agree with the comments. In the current version of the manuscript, the following revision was made:

“Such investigations may serve as a cost-effective epidemiological strategy to reduce the burden of NCDs, particularly within the Brazilian Unified Health System (Sistema Único de Saúde – SUS), the country’s publicly funded health system whose primary goal is to ensure universal access to comprehensive healthcare. Article 2 of Law No. 8.080/1990 establishes that ‘Health is a fundamental right of the human being, and it is the duty of the State to provide the necessary conditions for its full exercise’ [15].”

Reviewer #2: While the use of telephone interviews was appropriate given the constraints of the COVID-19 pandemic, it may have introduced potential information bias. This methodological limitation should be discussed more critically either in the Methods section or within the Discussion to acknowledge its impact on data validity.

Authors: We appreciate your suggestions and agree with the comments. In the current version of the manuscript, the following revision was made:

“This study presents some limitations. Among them, data collection was conducted through a telephone survey, a feasible method for epidemiological research involving large population groups; however, the responses may be influenced by recall bias [34]. Accordingly, the diagnosis of NCDs was obtained indirectly through self-report.”

Reviewer #2: There is significant repetition of numerical data already displayed in the tables. It would enhance the narrative to summarise the core findings rather than restating numbers verbatim.

Authors: We appreciate your suggestions and agree with the comments. In the current version of the manuscript, the following revision was made:

            “Participants who did not meet any of the 24-hour movement behavior recommendations had higher odds of hypertension (Table 2), diabetes (Table 3), and hypercholesterolemia (Table 4).

            A statistically significant association was also observed among individuals who met only the PA recommendation, as they had lower odds of cardiovascular disease in both the crude analysis and the adjusted model Likewise, participants who simultaneously met the PA and sleep recommendations had lower odds of cardiovascular disease in both the crude analysis and the adjusted model (Table 5).”

Reviewer #2: Considering the overwhelmingly female composition of the sample, it is recommended to perform subgroup or sensitivity analyses by sex to evaluate potential differences in the observed associations. 

Authors: We appreciate your suggestions and agree with the comments. We chose not to perform sex-stratified analyses due to the imbalance between groups, which could compromise the statistical stability of the results in the subgroups. However, we acknowledge this limitation and have addressed it in the discussion as a point to be considered in future studies with a more balanced sex distribution. In the current version of the manuscript, the following revision was made:

Another limitation of the study refers to the imbalance in the sample composition, which is predominantly female (92.9%). This considerable representation reduces the possibility of generalizing the findings to the overall adult Brazilian population, especially with regard to potential sex differences in the associations investigated.”

Reviewer #2: A more thorough discussion is needed regarding the gender imbalance in the sample and how this may limit the external validity or generalizability of the findings.

Authors: We appreciate your suggestions and agree with the comments. Indeed, we acknowledge that the sample in the present study is predominantly composed of women (92.9%), which limits the generalizability of the findings to the overall adult Brazilian population. This limitation has already been mentioned in the discussion section and will be more clearly emphasized, as suggested. In the current version of the manuscript, the following revision was made:

Another limitation of the study refers to the imbalance in the sample composition, which is predominantly female (92.9%). This considerable representation reduces the possibility of generalizing the findings to the overall adult Brazilian population, especially with regard to potential sex differences in the associations investigated.”

Reviewer #2:  The authors should also elaborate on the practical implications of their results, especially how these insights could inform public health strategies or interventions within the framework of Brazil's Unified Health System (SUS).

Authors: We appreciate the relevant consideration. Indeed, we agree that presenting the practical implications of the results is essential to enhance the applicability of the findings within the context of public health policies. Although the conclusion already addresses some suggestions, such as incorporating movement behaviors into awareness campaigns and primary care actions, we will revise and expand this section of the manuscript to more clearly highlight how the findings can inform strategies within the Brazilian Unified Health System (Sistema Único de Saúde – SUS). In the current version of the manuscript, the following revision was made: 

“Participants of the Health Academy Program in Brazil who did not meet any of the 24-hour movement behavior recommendations had higher odds of hypertension, diabetes, and multimorbidity. Additionally, noncompliance with the PA guideline among these participants was associated with increased odds of hypercholesterolemia. In contrast, those who adhered to the PA recommendation had lower odds of cardiovascular disease. These findings suggest that 24-hour movement behaviors may represent a low-cost epidemiological strategy for the prevention and control of NCDs within the Brazilian Unified Health System. The national health system could consider developing awareness campaigns emphasizing the importance of meeting PA recommendations, reducing screen time, and promoting adequate sleep duration and quality. In this regard, it is suggested that the 24-hour movement guidelines be integrated into primary care protocols through simple screening and counseling tools, such as questionnaires. Moreover, these recommendations may be incorporated into primary health care interventions to help prevent NCDs. Likewise, in clinical settings, health professionals could encourage discussions not only about PA but also about other movement behaviors and their implications for health. In addition, the training of SUS professionals to deliver educational approaches that integrate physical activity, sleep, and screen time is recommended. Leveraging existing programs, such as the Health Academy Program, may support the implementation of multicomponent and low-cost interventions. In this context, understanding the adverse effects of inadequate movement behaviors, whether individually or in combination, may help reduce healthcare costs and mortality associated with NCDs. Finally, future studies with longitudinal designs or experimental interventions may contribute to a better understanding of the effects of 24-hour movement behaviors on NCDs.”

Reviewer #2: The manuscript would be strengthened by suggesting directions for future studies, such as the inclusion of longitudinal research designs or experimental interventions to establish causality.

Authors: We appreciate your suggestions and agree with the comments. We fully agree that studies with longitudinal designs or experimental interventions are essential to establish causal relationships between 24-hour movement behaviors and noncommunicable diseases (NCDs). We have reinforced this recommendation by more clearly and objectively indicating that future studies should explore these methodological approaches in order to deepen the understanding of the underlying mechanisms. In the current version of the manuscript, the following revision was made:

“Finally, future studies with longitudinal designs or experimental interventions may contribute to a better understanding of the effects of 24-hour movement behaviors on NCDs.”

Reviewer #2: In reference number 1, the source is written as "WORLD HEALTH ORGANIZATION" in all capital letters. Unless it is a style requirement of the target journal, it is advisable to use sentence case (i.e., "World Health Organization") for consistency with other reference entries and standard citation practices. 

Authors: We appreciate your suggestions and agree with the comments. In the current version of the manuscript, the following change was made:

  1. “Word Health Organization. Non communicable diseases Available online: https://www.who.int/news-room/fact-sheets/detail/noncommunicable-diseases (accessed on 26 June 2025).”

Reviewer 3 Report

Comments and Suggestions for Authors

This article examines the relationship between adherence to 24-hour exercise behaviors and four common chronic diseases (hypertension, diabetes, hypercholesterolemia, and cardiovascular disease) among participants of the Health Academy Program in Brazil. The topic is highly relevant in public health, and the large sample size of the study enhances generalizability.

1) Model fit measures such as AUC, the Hosmer-Lemeshow test, and Nagelkerke R² are not provided in the logistic regression. This makes it difficult to assess the model's accuracy. Model fit measures should be provided below each table.

2) Information regarding interaction analyses (interaction, multicollinearity) should be provided in the model.

3) Important covariates such as smoking, alcohol consumption, and income level are missing. How might this affect the given model?

Author Response

Reviewer #3: This article examines the relationship between adherence to 24-hour exercise behaviors and four common chronic diseases (hypertension, diabetes, hypercholesterolemia, and cardiovascular disease) among participants of the Health Academy Program in Brazil. The topic is highly relevant in public health, and the large sample size of the study enhances generalizability.

Authors: We appreciate the reviewer’s positive comments regarding the relevance of the topic and the robustness of the sample. We are pleased to know that the study was considered pertinent to the field of public health.

Reviewer #3:  Model fit measures such as AUC, the Hosmer-Lemeshow test, and Nagelkerke R² are not provided in the logistic regression. This makes it difficult to assess the model's accuracy. Model fit measures should be provided below each table.

Authors: We appreciate your suggestions and agree with the comments. We recognize the importance of these statistics in evaluating the adequacy and accuracy of the models used. Following your suggestion, we have included two tables in Supplementary Material 1, presenting the goodness-of-fit measures for each model, including the area under the ROC curve (AUC), the Hosmer-Lemeshow test, Nagelkerke's R², and the Variance Inflation Factor (VIF). We considered the applicability of model quality analyses in relation to both binary and multinomial logistic regression. This information provides a more comprehensive view of the models' performance. In the current version of the manuscript, the following change was made:

“The goodness of fit of the binary logistic regression models was assessed using the area under the ROC curve (AUC), the Hosmer-Lemeshow test, and Nagelkerke’s R². In addition, the Variance Inflation Factor (VIF) was used to check for multicollinearity among the independent variables (Supplementary Material 1, Table 1).

The goodness-of-fit of the multinomial logistic regression models was assessed using Nagelkerke's R², and the Variance Inflation Factor (VIF) was used to detect potential multicollinearity issues among the independent variables (Supplementary Material 1, Table 2).”

Reviewer #3: Information regarding interaction analyses (interaction, multicollinearity) should be provided in the model.

Authors: We appreciate your suggestions and agree with the comments. We would like to inform you that multicollinearity among the independent variables was assessed using the Variance Inflation Factor (VIF), and this information is provided in Supplementary Material 1.

Reviewer #3: Important covariates such as smoking, alcohol consumption, and income level are missing. How might this affect the given model?

Authors: We appreciate your suggestions and agree with the comments. Indeed, covariates such as smoking, alcohol consumption, and income level are well known to be associated with both 24-hour movement behaviors and noncommunicable diseases, potentially acting as confounding factors in the investigated associations. However, these variables were not available in a complete or standardized format in the dataset used (Health Academy Program), which made their inclusion in the multivariate models unfeasible. The absence of information on smoking, alcohol consumption, and income represents another limitation of the study, as these variables may act as confounding factors, potentially affecting the magnitude of the observed associations. In other words, they may alter the true effect size estimated between 24-hour movement behaviors and health outcomes, which could lead to either an overestimation or underestimation of these associations. In the current version of the manuscript, the following change was made:

                The absence of data on smoking, alcohol consumption, and income represents a limitation, as these variables may act as confounding factors, influencing the estimated effects between 24-hour movement behaviors and health outcomes, which may result in either overestimation or underestimation of the observed associations.

Round 2

Reviewer 1 Report

Comments and Suggestions for Authors

Glad to see that the authors have been able to address almost all of the feedback. However, some of the changes shown in the main text were not included in the abstract. Specifically, Bonferroni adjusted p-values were not reported. I'd suggest at least mention what results were still significant after multiple comparison adjustments in the abstract.

Author Response

Reviewer #2: Glad to see that the authors have been able to address almost all of the feedback. However, some of the changes shown in the main text were not included in the abstract. Specifically, Bonferroni adjusted p-values were not reported. I'd suggest at least mention what results were still significant after multiple comparison adjustments in the abstract.

Authors: We appreciate your suggestions and agree with the comments. In the current version of the manuscript, the following change was made:

All of these results remained significant after adjustments for multiple comparisons.